# Realistic Advantages of Early Surgical Drain Removal after Pancreatoduodenectomy: A Single-Institution Retrospective Study

**DOI:** 10.3390/jcm10122716

**Published:** 2021-06-19

**Authors:** So-Jeong Yoon, So-Kyung Yoon, Ji-Hye Jung, In-Woong Han, Dong-Wook Choi, Jin-Seok Heo, Sang-Hyun Shin

**Affiliations:** Division of Hepatobiliary-Pancreatic Surgery, Department of Surgery, Samsung Medical Center, Sungkyunkwan University School of Medicine, Seoul 06351, Korea; wooyabi@gmail.com (S.-J.Y.); bluesona8@gmail.com (S.-K.Y.); sog-hei@hanmail.net (J.-H.J.); cardioman76@gmail.com (I.-W.H.); dwchoi1218@gmail.com (D.-W.C.); jsheo.md@gmail.com (J.-S.H.)

**Keywords:** pancreatoduodenectomy, postoperative pancreatic fistula, surgical drain, perianastomotic drain, enhanced recovery after surgery

## Abstract

The latest guidelines from the Enhanced Recovery After Surgery (ERAS^®^) Society stated that early drain removal after pancreatoduodenectomy (PD) is beneficial in decreasing complications including postoperative pancreatic fistulas (POPFs). This study aimed to ascertain the actual benefits of early drain removal after PD. The data of 450 patients who underwent PD between 2018 and 2020 were retrospectively reviewed. The surgical outcomes were compared between patients whose drains were removed within 3 postoperative days (early removal group) and after 5 days (late removal group). Logistic regression analysis was performed to identify the risk factors for clinically relevant POPFs (CR-POPFs). Among the patients with drain fluid amylase < 5000 IU on the first postoperative day, the early removal group had fewer complications and shorter hospital stays than the late removal group (30.9% vs. 54.5%, *p* < 0.001; 9.8 vs. 12.5 days, *p* = 0.030, respectively). The incidences of specific complications including CR-POPFs were comparable between the two groups. Risk factor analysis showed that early drain removal did not increase CR-POPFs (*p* = 0.163). Although early drain removal has not been identified as apparently beneficial, this study showed that it may contribute to an early return to normal life without increasing complications.

## 1. Introduction

Pancreatoduodenectomy (PD) is a major surgical procedure, mostly performed to remove periampullary tumors arising from the head of the pancreas, distal bile duct, duodenum, or ampulla of Vater. The procedure is associated with perioperative mortality rates up to 2% [1], and overall complication rates of 40 to 50%, even at high-volume centers [2,3]. One of the most disturbing complications is postoperative pancreatic fistulas (POPFs) and particularly, clinically relevant postoperative pancreatic fistulas (CR-POPFs) are related to increased postoperative hemorrhage, severe infectious complications, and mortality with prolonged hospital stays and expenses [4,5]. There have been many attempts to predict and prevent POPFs, but the reported incidence is still high [6,7,8].

To reduce perioperative stress and optimize recovery after surgery, the Enhanced Recovery After Surgery (ERAS^®^) guidelines for PD were first introduced in 2012 and updated in 2019 [9]. Regarding perianastomotic drainage, the guidelines strongly recommend early drain removal at postoperative 72 h in patients with drain fluid amylase (DFA) levels of <5000 IU/L on the first postoperative day (POD1). This management was found to significantly decrease morbidity including POPFs and hospital stays in several prospective studies [10,11,12,13].

However, according to a recent research survey investigating the application of the ERAS guidelines by Korean hepato-biliary-pancreatic surgeons [14], only 13.3% of the participants were following the recommendations for drain removal. The author suggested that one of the reasons for the low acceptance rate was the large discrepancy between the guidelines and traditional experience-based management, in which drains are removed at surgeons’ discretion with the aid of serial DFA results or follow-up imaging.

The clinical pathway for pancreatoduodenectomy in our institute, which is one of the largest tertiary referral cancer centers in South Korea, still adheres to the traditional experience-based management in terms of drain removal. To promote evidence-based management, we implemented an early removal protocol over the past two years. Therefore, the purpose of the present study was to report a single-center experience with drain management conducted during the transitional period and demonstrate the realistic advantages of early drain removal in terms of postoperative outcomes.

## 2. Materials and Methods

From September 2018 to July 2020, a total of 450 patients underwent pancreatoduodenectomy in Samsung Medical Center. The medical records of the patients, including clinical, pathological, and surgical outcomes, were retrospectively reviewed. This study was approved by the Institutional Review Board of Samsung Medical Center (Seoul, Korea, approval no. 2020-09-122). 

The data on preoperative biliary drainage and neoadjuvant treatment were collected. For intraoperative data, operating times, blood transfusion status, pancreatic texture, and the size of the main pancreatic duct were reviewed. The location of the tumors was noted in the final pathology reports.

In all patients with pancreatoduodenectomy, two or three intra-abdominal drains were placed, with at least one of them near the pancreatojejunostomy. The quality of the drains and DFA levels were recorded, starting from POD1. Follow-up computed tomography (CT) scans were performed after POD5 to detect any intra-abdominal complications such as anastomosis dehiscence or fluid collection. Before 2018, the drains were removed when there was no evidence of POPF on CT scans. Since 2018, the early drain removal protocol has been standardized, and drain removal is considered on POD3 in patients with POD1 DFA < 5000 IU. For postoperative management other than drain removal, all surgeons followed the clinical pathway of the institution. Patients were discharged when they could tolerate a regular diet and there was no sign of infectious complication. 

In the comparison of postoperative outcomes, POD1 DFA levels, the timing of drain removal, in-hospital complications, and the 90-day operation-related readmission rate were included. The Clavien–Dindo classification (CD classification) was applied for evaluating the severity of surgical complications. POPF was counted separately from CD classification since it is a pancreatectomy-specific outcome. According to the International Study Group of Pancreatic Fistula (ISGPF) definition and grading system [15], POPF was diagnosed when the DFA was greater than three times the upper normal serum value starting from POD3 and classified into three different grades of biochemical leaks (BLs), and grades B and C (also as CR-POPFs). In patients with POPFs or intra-abdominal fluid collection detected on postoperative CT, percutaneous or endoscopic drainage was performed.

The comparison of the clinical characteristics between the different groups of patients was performed using Student’s *t*-test, chi-squared test, and Fisher’s exact test. Binary logistic regression was used for identifying the risk factors for CR-POPFs, and odds ratios (ORs) were reported with 95% confidence intervals (CIs). Variables with *p*-values of less than 0.05 were regarded as statistically significant. All statistical analyses were performed using IBM SPSS software (version 26, SPSS Inc., Chicago, IL, USA).

## 3. Results

### 3.1. Demographics

The demographic and clinicopathologic profiles of the patients are shown in Table 1. Surgical resections were performed for 233 (49.6%) patients with pancreatic tumors and 227 (50.4%) patients with other periampullary tumors arising from the distal common bile duct, duodenum, or ampulla of Vater. The abdominal drains were removed within the first 3 postoperative days in 91 (20.2%) patients (early removal group), and more than 5 days after surgery in 359 (79.8%) patients (late removal group).

### 3.2. The Comparison between the Early Removal Group and the Late Removal Group

#### 3.2.1. In All Included Patients (*n* = 450)

A comparison of the clinical factors and surgical outcomes was conducted between the early and late removal groups in all patients (Table 2). No statistically significant difference was observed in preoperative clinical variables between the two groups. The mean operating time was longer in the early removal group than in the late removal group (327.6 min vs. 306.5 min, *p* = 0.009).

In terms of surgical outcomes, the overall complication rate was higher in the late removal group (30.8% vs. 58.8%, *p* < 0.001), although the incidence of CD grade III or higher complications was comparable to that of the early removal group. The early removal group had significantly lower rates of POPFs and CR-POPFs than the late removal group (41.8% vs. 60.2%, *p* = 0.002; 6.6% vs. 16.7%, *p* = 0.015, respectively). The rate of additional drainage tube insertion after drain removal did not differ between the two groups. The mean postoperative hospital stay was 9.8 days in the early removal group and 13.3 days in the late removal group (*p* < 0.001). There was no significant difference in readmission rates between the two groups.

#### 3.2.2. In Patients with POD1 DFA < 5000 IU (*n* = 338)

Among 338 patients with POD1 DFA levels of less than 5000 IU, 81 (23.9%) patients belonged to the early removal group and 257 (76.1%) patients to the late removal group (Table 3). The pre- and intraoperative factors did not differ significantly.

The overall complication rate was higher in the late removal group, whereas the rate of CD grade III or higher complications was not significantly different between the groups (67.6% vs. 84.8%, *p* < 0.001; 17.3% vs. 22.2%, *p* = 0.346, respectively). POPFs occurred in 35.8% of the early removal group, which was lower than that in the late removal group (48.2%, *p* = 0.050). However, the incidence of CR-POPFs or additional drainage was not significantly different. The patients in the early removal group had shorter hospital stays than the patients in the late removal group (9.8 days vs. 12.5 days, *p* = 0.030). No significant difference was found regarding the readmission rate between the two groups.

### 3.3. Risk Factor Analysis for CR-POPF

For the above-mentioned group of patients with POD1 DFA levels of less than 5000 IU, risk factor analysis for CR-POPFs was performed (Table 4). In the univariable analysis, soft pancreatic texture (OR = 2.261, 95% CI: 1.106–4.622, *p* = 0.025) and the tumor origin (OR = 0.212, 95% CI: 0.093–0.484, *p* < 0.001) were associated with the development of CR-POPFs. In the multivariable analysis, tumors arising from the pancreas were found to be a protective factor for CR-POPFs (OR = 0.267, 95% CI: 0.113–0.629, *p* = 0.003). The timing of drain removal was not a significant risk factor for CR-POPFs, both in univariable and multivariable analyses.

## 4. Discussion

It has been more than 10 years since the timing of surgical drain removal emerged as a topic of active debate, and the current ERAS recommendations include the removal of drains on POD3 in patients with DFA levels of <5000 IU/L on POD1 [9]. Several previous studies have demonstrated the feasibility and benefits of early drain removal after PD in low-risk patients [11,13,16], and we are aware of its advantages. However, a former national survey investigating the application of individual items in the ERAS guidelines showed that many surgeons adopt only some of those items, and many still rely on inconsistent experience-based management for other items including drain removal [14]. A recent study of the Japanese Society of Surgical Metabolism and Nutrition conducted by Kaibori et al. [17] showed encouraging results; their promotion project improved the rate of implementation of the ERAS protocol. However, the report did not indicate the degree of improvement in the detailed items. Therefore, in the present study, we aimed to record the results during the past two transitional period years when the early drain removal protocol was implemented in our institute.

In our database, there were only 91 (20.2%) patients in the cohort whose drains were removed within POD3. Even though the early removal protocol has been implemented in the institute since 2018, not all surgeons immediately followed the guidelines for fear of adverse events such as failing to notice anastomotic leakage. In accordance with several studies investigating the adherence rates to the ERAS items [14,18,19], adherence to postoperative items including drain management tended to be lower than that to the preoperative items. This could be explained by its relevance to complications in addition to the drastic difference from traditional management. In this regard, we investigated the rate of additional percutaneous or endoscopic drainage tube insertion for intra-abdominal fluid collection or POPFs after drain removal, and there was no statistical difference between the early and late removal groups. This suggests that early drain removal does not increase adverse events requiring additional intervention. In addition, it should be noted that drains can help detect intra-abdominal complications, but they do not fundamentally prevent them. On that basis, there is a need to consider a more extensive implementation of the early drain removal protocol.

With regard to CR-POPFs, many studies have analyzed the risk factors and proposed risk scoring systems [7,20,21,22,23,24]. In addition, there have been attempts to set criteria for the early removal of drains [12,25]. Among several variables, the most emphasized was the POD1 DFA level. To identify other factors independent of POD1 DFA levels, we performed multivariable risk factor analysis in patients with POD1 DFA levels < 5000 IU, and tumor location was found to be an independent factor. However, the timing of drain removal did not increase the risk of CR-POPFs. This implies that drains can be safely removed earlier in patients with POD1 DFA levels of <5000 IU, without increasing the risk of intra-abdominal complications, including POPFs, while reducing the length of hospital stays and enhancing early recovery. Meanwhile, in the analysis of all 450 patients including those with POD1 DFA levels of ≥5000 IU, early drain removal had the advantages of lower complication rates, including POPFs, and shorter hospital stays over late drain removal. Altogether, further studies on the risk factors for POPFs or postoperative intra-abdominal complications, other than DFA levels, are necessary to select the candidates for early and safe removal of surgical drains.

There were several limitations to this study. Above all, this was a single-center retrospective study, which is prone to selection bias. Information bias is also of concern because the data on postoperative events such as complications were collected from previously archived medical records. Secondly, regardless of the clinical pathway, which was modified in 2018 according to the ERAS guidelines, each surgeon applied the new drain protocol at different times. During the transition period, not all surgeons removed drains in POD3 patients with POD1 DFA levels <5000 IU, and the definition of early or late drain removal was unclear. The timing of follow-up imaging and the date of discharge also varied. Therefore, the influence of surgeon-specific factors on operative and post-operative outcomes cannot be excluded. Nevertheless, based on the results of our study, which included a relatively large number of patients undergoing PD, all surgeons in our institute are now considering the practical implementation of early drain removal.

## 5. Conclusions

In conclusion, we investigated the realistic advantages of early drain removal after PD and found that the evidence-based protocol for early drain removal did not increase postoperative morbidity and may reduce the length of hospital stay.

## Figures and Tables

**Table 1 jcm-10-02716-t001:** Demographics and clinical characteristics, all patients (*n* = 450).

Variables	*n* (%) or Mean (±SD)	Variables	*n* (%) or Mean (±SD)
Age, mean	64.9 (±9.98)	Pathology	
Sex		Pancreatic tumor	223 (49.6%)
Male	253 (56.2%)	Others	227 (50.4%)
Female	197 (43.8%)	POD1 DFA	
BMI (kg/m^2^)	23.7 (±3.15)	<5000 IU	338 (75.1%)
ASA score		≥5000 IU	112 (24.9%)
I	32 (7.1%)	Drain removal	
II	339 (75.3%)	Early (within 3 days)	91 (20.2%)
III	78 (17.4%)	Late (after 5 days)	359 (79.8%)
IV	1 (0.2%)	Overall Complications	
Preop. Biliary drainage		No	211 (46.9%)
No	186 (41.3%)	Yes	239 (53.1%)
Yes	264 (58.7%)	C-D classification	
Neoadjuvant therapy		<Grade III	365 (81.1%)
No	401 (89.1%)	≥Grade III	85 (18.9%)
Yes	49 (10.9%)	POPF	
Operation time (min)	310.8 (±63.71)	No	196 (43.5%)
Pancreatic texture		BCL	188 (41.8%)
Soft	192 (42.7%)	Grade B	63 (14.0%)
Moderate	153 (34.0%)	Grade C	3 (0.7%)
Hard	101 (22.4%)	90-day mortality	0 (0%)
N/A	4 (0.9%)		
Intraop. transfusion			
No	430 (95.6%)		
Yes	20 (4.4%)		
MPD size (mm)	3.57 (±2.09)		

SD, standard deviation; BMI, body mass index; ASA, American Society of Anesthesiologists; Preop., preoperative; N/A, not available; Intraop., intraoperative; MPD, main pancreatic duct; POD, postoperative day; DFA, drain fluid amylase; C-D, Clavien–Dindo; POPF, postoperative pancreatic fistula; BCL, biochemical leakage.

**Table 2 jcm-10-02716-t002:** Comparison of the early removal group and the late removal group in all patients (*n* = 450).

Variables	Early Removal (*n* = 91)	Late Removal (*n* = 359)	*p*
*Clinicopathologic factors*			
Age, mean	64.9	65.0	0.930
Sex			0.364
Male	55 (60.4%)	198 (55.2%)	
Female	36 (39.6%)	161 (44.8%)	
BMI (kg/m^2^), mean	23.4	23.8	0.361
ASA score			0.532
I–II	73 (80.2%)	298 (83.0%)	
III–IV	18 (19.8%)	61 (17.0%)	
Preop. Biliary drainage, Yes	53 (58.2%)	211 (58.8%)	0.927
Neoadjuvant therapy, Yes	8 (8.8%)	41 (11.4%)	0.472
Operation time (min), mean	327.6	306.5	0.009
Pancreatic texture			0.066
Soft	31 (34.4%)	161 (45.2%)	
Moderate	40 (44.4%)	113 (31.7%)	
Hard	19 (21.2%)	82 (23.1%)	
MPD size (mm), mean	3.5	3.6	0.566
Intraop. Transfusion, Yes	2 (2.2%)	18 (5.0%)	0.392
Pathology			0.064
Pancreatic tumors	53 (58.2%)	170 (47.4%)	
Others	38 (41.8%)	189 (52.6%)	
*Surgical outcomes*			
Overall complications, Yes	28 (30.8%)	211 (58.8%)	<0.001
C-D grade ≥ III, Yes	12 (13.2%)	73 (20.3%)	0.120
POPF, Yes	38 (41.8%)	216 (60.2%)	0.002
CR-POPF, Yes	6 (6.6%)	60 (16.7%)	0.015
Additional drainage *	5 (5.5%)	16 (4.5%)	0.590
Length of stay (days)	9.8	13.3	<0.001
Re-admission	6 (6.6%)	39 (10.9%)	0.225

POD, postoperative day; DFA, drain fluid amylase; BMI, body mass index; ASA, American Society of Anesthesiologists; Preop., preoperative; MPD, main pancreatic duct; Intraop., intraoperative; C-D, Clavien–Dindo; CR-POPF, clinically relevant postoperative pancreatic fistula. * Additional drainage tube inserted by percutaneous or endoscopic approach after removal of surgical drains.

**Table 3 jcm-10-02716-t003:** Comparison of the early removal group and the late removal group in patients with POD1 DFA < 5000 IU (*n* = 338).

Variables	Early Removal (*n* = 81)	Late Removal (*n* = 257)	*p*
*Clinicopathologic factors*			
Age, mean	65.3	65.6	0.836
Sex			0.160
Male	51 (63.0%)	139 (54.1%)	
Female	30 (37.0%)	118 (45.9%)	
BMI (kg/m^2^), mean	23.5	23.5	0.973
ASA score			0.532
I–II	65 (80.2%)	214 (83.3%)	
III–IV	16 (19.8%)	43 (16.7%)	
Preop. Biliary drainage, Yes	50 (61.7%)	155 (60.3%)	0.820
Neoadjuvant therapy, Yes	8 (9.9%)	37 (14.4%)	0.296
Operation time (min), mean	322.6	307.1	0.051
Pancreatic texture			0.184
Soft	25 (31.3%)	93 (36.5%)	
Moderate	36 (45.0%)	86 (33.7%)	
Hard	19 (23.8%)	76 (39.8%)	
MPD size (mm), mean	3.5	3.9	0.183
Intraop. Transfusion, Yes	2 (2.5%)	15 (5.8%)	0.380
Pathology			0.450
Pancreatic tumors	48 (59.3%)	140 (54.5%)	
Others	33 (40.7%)	117 (45.5%)	
*Surgical outcomes*			
Overall complications, Yes	117 (67.6%)	140 (84.8%)	<0.001
C-D grade ≥ III, Yes	12 (14.8%)	47 (18.3%)	0.473
POPF, Yes	29 (35.8%)	124 (48.2%)	0.050
CR-POPF, Yes	4 (4.9%)	30 (11.7%)	0.079
Additional drainage *	4 (4.9%)	6 (2.3%)	0.259
Length of stay (days)	9.8	12.5	0.030
Re-admission	5 (6.2%)	23 (8.9%)	0.429

POD, postoperative day; DFA, drain fluid amylase; BMI, body mass index; ASA, American Society of Anesthesiologists; Preop., preoperative; MPD, main pancreatic duct; Intraop., intraoperative; C-D, Clavien–Dindo; CR-POPF, clinically relevant postoperative pancreatic fistula. * Additional drainage tube, inserted by percutaneous or endoscopic approach after removal of surgical drains.

**Table 4 jcm-10-02716-t004:** Binary logistic regression analysis for CR-POPFs in patients with POD1 DFA < 5000 IU (*n* = 338).

	Univariable Analysis			Multivariable Analysis		
Variable	OR	95% CI	*p*	OR	95% CI	*p*
Age	1.007	0.970–1.046	0.712			
Sex, female	0.888	0.432–1.824	0.746			
BMI	1.092	0.982–1.213	0.103			
ASA score	1.039	0.504–2.138	0.918			
Preop. biliary drainage	1.914	0.864–4.240	0.110			
Pancreatic texture, soft	2.261	1.106–4.622	0.025	1.834	0.782–4.305	0.163
Intraop. transfusion	2.005	0.546–7.361	0.295			
Pathology, pancreatic tumors	0.212	0.093–0.484	<0.001	0.267	0.113–0.629	0.003
Early drain removal	0.393	0.134–1.151	0.089	0.330	0.075–1.459	0.144

CR-POPF, clinically relevant postoperative pancreatic fistula; POD, postoperative day; DFA, drain fluid amylase; BMI, body mass index; ASA, American Society of Anesthesiologists; Preop., preoperative; Intraop., intraoperative; OR, odds ratio; CI, confidence interval.

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
