# Peer review of "Realistic Advantages of Early Surgical Drain Removal after Pancreatoduodenectomy: A Single-Institution Retrospective Study"

_jcm, 2021, doi:10.3390/jcm10122716_

Round 1
Reviewer 1 Report
Dear authors,
first of all I want to thank you for the opportunity to review your interesting work. You presented a large cohort of pancreaticoduodenectomies, the aim of your work is clear and the result interesting and well stated. Unfortunately as it is, in my opinion, can raise some concern that have to be asses:
- Major revisions: as you correctly mentioned the fact that despite the introduction of the protocol still the 76.1% of patients didn't underwent the early removal of the drains with POD1 DFA < 5000. This clearly represent the fact that only a minority of the surgeons in your hospital comply with the protocol. Moreover can introduce a selection bias, here you could be not comparing the usefulness of the early removal but simply the outcomes of patients treated by different surgeons. Additionally one group of this surgeons could be expected to be more found with ERAS predicament and so a shorted length of stay could be related to the overall management of the patients and not only the removal of the drains. In my opinion you have to assess this point introducing in the analysis the operating/managing surgeons.
- In the "all included" cohort is peculiar that the incidence of CD grade III is not different and the CR-POPF rate is different among the groups. Usually in pancreatic surgery this two entity overlap in in majority of cases. Could you provide a categorization of the type of CD grade III or perform an analysis on the reoperation rate? Also a comment on this findings could be needed.
- Did you apply at the time of the study a structured ERAS protocol in your institution with discharge criteria defined a priori (or only the removal strategy)? If yes please provide it or state it. If not if should be discuss as a weak point.
Minor revision: is very unclear in the methods section how you manage the postoperative period in the early removal group, did you perform the CT scan also in all this patients? If yes what was the timing? Please provide more information on your postoperative management, this also could be a bias.
Best regards
Author Response
We sincerely appreciate your comments regarding our manuscript and the opportunity you gave us for a revision. We have written our responses to your comments and the revisions that have been made accordingly
- The post-operative management of each surgeon in our institution.
Except for surgical drain management, the surgeons in our institution followed the same clinical pathway. For example, fluid management, the timing of follow-up laboratory tests and CT scans were performed without discrimination. Also, the patients were discharged when they could tolerate a regular diet and there was no sign of other complications such as infections. We added this in the 3rd paragraph of the methods section.
- About the complications more severe than CD grade III
We agree that you had a point. So we distinguished POPF from complication ≥ CD grade III in the analysis and modified the results in Table 1, 2, and 3. We additionally mentioned that in the 4th paragraph of the methods. In this case, complications ≥ CD grade III included hemorrhage, intra-abdominal fluid collection not related to POPF, bile fistula and biliary stricture, which needed endoscopic or radiologic intervention.
- About other components of the ERAS guidelines.
As mentioned before, all surgeons are following the same clinical pathway except for the drain management. In terms of our clinical pathway, although it does not contain all the components of the ERAS guidelines, it includes some of the major components which are strongly recommended. For example, antibiotic prophylaxis, postoperative glycemic control, and fluid management correspond to the guidelines.
- About follow-up CT scans in the patients.
The patients in the early removal group also took CT scans at POD5, as mentioned in 3rd paragraph of the methods. Since the former description might make readers confused, we corrected the sentence more concretely.
Please see the attachment. Thank you.
Reviewer 2 Report
We have to congratulate the authors for this nice study highlighting the need of early drain removal after PD in patients with low risk of POPF and a drain fluid amaylase level of <5000UI at day 1. More RCTs are then needed to confirm the conclusions of the authors.
I have one issue that need to be adressed more clearly. Did the authors use the ERAS programm in their patients after PD? This should be clearly defined and described in the method and result sections.
Author Response
We are very grateful for your review. About the postoperative management of our institution, we added a detailed description of our clinical pathway in the 3rd and 4th paragraphs of the methods. Except for surgical drain management, the surgeons in our institution followed the same clinical pathway. For example, fluid management, the timing of follow-up laboratory tests and CT scans were performed without discrimination. Also, the patients were discharged when they could tolerate a regular diet and there was no sign of other complications such as infections.
Please see the attached manuscript which was revised. Thank you.
Reviewer 3 Report
Thank you for the opportunity to review this paper. The authors reported a retrospective review of an impressive number of 450 pancreatoduodenectomy performed between January 2014 and March 2019, from one of the largest tertiary referral center in South Korea.
The aim of this study was to evaluate the feasibility and safety of early drain removal at POD3 after PD. The authors concluded that early drain removal did not increase postoperative morbidity and may reduce the length of hospital stay.
I read with great interest this work of an experienced team, during a recent period, on a crucial point: the close relation between early drain removal and pancreatic fistula. Even though this is a well-written paper on a useful question, there are several confusing points and some elements can be improved in this article:
Major revisions:
- Special focus is done on the subgroup analysis of the 81 patients with DFA <5000 IU/L on POD1 and with low risk of CR-POPF. Thus, multivariate analysis on CR-POPF in this subpopulation will probably be affected by the statistical power and could result in a type II error. Percutaneous and endoscopic drainage are 5 vs 2.8%, even not statistically significant.
- Furthermore, I could be interesting to have a look on the 10 patients with early drain removal with DFA>5000 IU/L even if it was not the aim of the study.
- On the methods section, substantial change have to be made:
1/ Justification for early and late drainage removal, and why there is no patient between 3 and 5 days?
2/ A precise description of the drainage
3/ A note about the mitigation strategy used
- On the result section, focus could be done on the delay of CR-POPF appearance (between POD3 and later).
Minor revisions:
I have underlined four points to improve the methodology, the results and the discussion.
- In the introduction section, mortality for PD is often more than 2% even in high-volume center, mostly depending of the pancreatic parenchyma texture and the need for vascular resection. Please add references on that. Furthermore, the most up-to-date score is the ua-FRS and should be cited.
- In the methods section, please define and cite reference for CR-POPF.
- In the results section, table 2, please add the 30- and 90-days mortality to have complete description of the whole population of PD
- Three levels of parenchyma texture are often difficult to evaluate and reproduce, it is simpler and more relevant to define soft and normal parenchyma texture (10.1097/SLA.0000000000004855)
For all these reasons, I recommend major revision but once again, it is a well-wrote study on a topic of debate in modern era.
Author Response
We really appreciate your thorough review and we tried to cover the points you made for the improved manuscript.
- About those 10 patients whose drains were removed earlier regardless of the higher DFA level (>5000IU) on POD#1.
Regardless of POD#1DFA, there were some patients with early drain removal at surgeons’ discretion. Since the data were retrospectively reviewed, we could not standardize all the details. We guess this might not be the main point of our study, so we would just provide the data of those 10 patients for you, here in the response.
Among those with POD1DFA>5000IU (n=112), 10 patients were in the early removal group and 102 patients were in the late removal group. The rates of POPF and CR-POPF were comparable between the two groups (90.0 % vs 90.2%, p=0.661; 20.0% vs 29.4%, p=0.722, respectively).
- Drains removed between POD3 and POD5
There were only two ways of drain management in our institution during the study period.
First, early drain removal at POD3, and second, late drain removal after follow-up CT scans, which were mostly done after POD5. This is why there was no patients whose drains were removed between POD3 and POD5.
For better understanding, we added some details about postoperative management in 3rd paragraph of the methods.
- A precise description of the drains
Two or three intra-abdominal drains were placed, with at least one of them near the pan-creatojejunostomy. This sentence was added in the 3rd paragraph of the methods.
- About the delayed revelation of POPF
We defined POPF as DFA > 3 times upper normal limit after POD3, according to the ISGPF definition. Also, the patients who were identified with POPF on the postoperative CT or who had to be re-admitted for sepsis from POPF were also included in the analyses.
- Reference for mortality rate after pancreatoduodenectomy
The appropriate reference was cited after the related sentence in the introduction.
- Reference for ua-FRS
The updated alternative fistula risk score was additionally cited in the 1st paragraph of the introduction, as you pointed out.
- Reference for ISGPF definition and grade
The citation was added in the 4th paragraph of the methods.
- About 30-day and 90-day mortality
Since there was no mortality case (both in 30-day and 90-day) during the study period, we did not perform any analysis on it. But for the potential readers, we added ’90-day mortality’ in Table 1.
- About the pancreatic texture
We totally agree that this variable could not be objectively measured. In the operation record of our institution, the pancreatic texture is classified into these three groups: soft, moderate, hard. This is the reason we analyzed the texture with three categories. We will consider narrowing down the categories into two groups for the future studies.
Please see the attached manuscript which was revised. Thank you.
Round 2
Reviewer 1 Report
Dear authors thank you for your kind and exaustive answers. We hope you are also planning a prospective evaluation now that the protocol will be completely implemented in your istitution. Best regards